# Src Family Kinases: A Potential Therapeutic Target for Acute Kidney Injury

**DOI:** 10.3390/biom12070984

**Published:** 2022-07-14

**Authors:** Nannan Li, Guoxin Lin, Hao Zhang, Jian Sun, Ming Gui, Yan Liu, Wei Li, Jishi Liu, Juan Tang

**Affiliations:** 1Department of Nephrology, The Third Xiangya Hospital, Central South University, Changsha 410013, China; 208311024@csu.edu.cn (N.L.); zhanghaoliaoqing@163.com (H.Z.); sunjian105@sina.com (J.S.); zkbgm@126.com (M.G.); nianbiecabell@sina.com (Y.L.); liweiss2008@126.com (W.L.); 2Department of Anesthesiology, The Third Xiangya Hospital, Central South University, Changsha 410013, China; lgx_mzk@126.com

**Keywords:** Src family kinases, acute kidney injury, inflammation, oxidative stress, ER stress, autophagy, fibrosis

## Abstract

Src family kinases (SFKs) are non-receptor tyrosine kinases and play a key role in regulating signal transduction. The mechanism of SFKs in various tumors has been widely studied, and there are more and more studies on its role in the kidney. Acute kidney injury (AKI) is a disease with complex pathogenesis, including oxidative stress (OS), inflammation, endoplasmic reticulum (ER) stress, autophagy, and apoptosis. In addition, fibrosis has a significant impact on the progression of AKI to developing chronic kidney disease (CKD). The mortality rate of this disease is very high, and there is no effective treatment drug at present. In recent years, some studies have found that SFKs, especially Src, Fyn, and Lyn, are involved in the pathogenesis of AKI. In this paper, the structure, function, and role of SFKs in AKI are discussed. SFKs play a crucial role in the occurrence and development of AKI, making them promising molecular targets for the treatment of AKI.

## 1. Introduction

Acute kidney injury (AKI) is defined by a rapid increase in serum creatinine (≥1.5 times the baseline within 7 days) or a rapid decrease in urine volume (<0.5 mL/kg/h for 6 h) [1]. Regardless of the economic situation of countries and regions, AKI is a common disease in all countries of the world, with high morbidity, high mortality, and high medical care costs [2]. After recovery from AKI, patients still carry the risk of developing chronic kidney disease (CKD), end-stage renal disease (ESKD), and death [3]. Apart from that, the incidence of AKI is mounting, and the impact of AKI can cause far-reaching consequences on long-term health, and the cost can be far heavier than before [4].

AKI is usually caused by ischemia-reperfusion injury, septicemia, and drug toxicity. Various toxic or ischemic injuries propagate renal tubular injury, which can be mediated by microvascular dysfunction, oxidative stress, endoplasmic reticulum stress, autophagy, immune disorder [5], inflammation, and maladaptive repair [6]. Microcirculation is impaired, resulting in an imbalance of supply and use of NO, ROS, and O_2_, followed by hypoxia and oxidative stress [6]. Long-term hypoxia [7], microvascular damage, mitochondrial disturbance [8], and inflammation lead to the transition of AKI to CKD [6]. Maladaptive repair terminates in kidney fibrosis, which further aggravates the outcome of renal fibrosis [9].

At present, the treatment for AKI mainly includes eliminating the etiology, actively preventing and treating complications, and supporting treatment [10]. The treatment of AKI is difficult because of the lack of drugs aimed at the pathogenesis and target of AKI [6]. The Src kinase family (SFKs), a non-receptor tyrosine kinase family with 11 members [11], is widely studied in the kidney. Recently, significant attention has been given to the field of the role of SFKs in AKI [12,13]. Therefore, SFKs are critical for the pathogenesis of AKI, which are expected to provide promising and new therapeutic interventions in the near future.

## 2. Overview of SFKs

### 2.1. Introduction and Structure of SFKs

Tyrosine kinase is an enzyme that acts by phosphorylating tyrosine residues of target proteins. Given the tyrosine kinases’ location in cells, they can be categorized as receptor tyrosine kinases (RTK) and non-receptor tyrosine kinases. Src family kinases (SFKs) belong to non-receptor protein tyrosine kinases, which play a vital role in multiple processes of cell activity, such as cell growth, division, migration, and survival signaling pathways [14].

SFKs are composed of 11 members in humans, including Src, Fyn, Yes, Blk, Brk (also known as PTK6), Frk (also known as Rak), Fgr, Hck, Lck, Srms (also called Srm), and Lyn [11,15,16,17]. These kinases can be divided into two subfamilies, the SrcA subfamily (Src, Yes, Fyn and Fgr) and the Lyn-related SrcB subfamily (Lyn, Hck, Lck and Blk) [18,19]. In addition, there are three SFK-related kinases (the Brk family), namely Brk, Frk, and Srm [19], which lack N-terminal myristoylation/palmitoylation sites, a structure common to all other SFKs family members [16,19,20]. Src, Fyn and Yes are expressed in almost all cell types [21]. On the contrary, Blk, Fgr, Hck, Lck, and Lyn are mainly found in hematopoietic cells [15,22]. Srms was first found in mouse neural precursor cells [23] and is widely expressed in normal mammalian tissue samples [15,20]. Brk is mainly expressed in epithelial cells of the gastrointestinal tract, skin, and prostate [20,24]. Frk mainly occurs in the kidney and liver [25,26] but is widely expressed in many tissues [25,26,27].

All SFKs members have similar structures (Figure 1), including an N-terminal 14-carbon myristoyl motif, Src homology domain 4 (SH4), a unique domain, SH3 domain, SH2 domain, SH1 (catalytic domain), and a C-terminal regulatory tail [28,29]. Among them, the SH4 domain is a region containing from 15 to 17 amino acid residues that are involved in anchoring proteins to membranes via myristoylation or palmitoylation [30,31]. In addition, the interaction between the SH4 domain and αF pocket in the C-terminal domain enhances the self-inhibition mediated by SH2/SH3 domain and regulates kinase activity [32]. Different from other domains, the amino acid sequence of the unique domain is not conserved in SFKs [17], which facilitates each family member to interact with specific receptors or proteins [17,30]. The SH2 domain is necessary for mediating phosphotyrosine-dependent protein–protein interactions [33], while the SH3 domain mediates molecular and intermolecular interactions by binding to proline-rich regions [34,35]. SH1 contains a catalytic kinase domain, and the full catalytic activity of SFKs requires autophosphorylation at Tyr416 (according to chicken numbering), which is activated by various transmembrane receptor proteins (including receptor tyrosine kinase, G protein-coupled receptor, integrin, and cytokine receptor) [36,37,38]. The phosphorylation of Tyr527 at C-teminal inactivates Src kinase by inhibiting autophosphorylation of Tyr416 in the catalytic domain [11]. Tyr527 is phosphorylated by Csk (C-terminal Src kinase) or Chk (Csk homologous kinase) [11,22]. The interaction between phosphorylated Tyr527 and the SH2 domain is helpful in maintaining the inactive state of SFKs [15,31]. Therefore, the dephosphorylation of phosphorylated Tyr527 is of great importance for the activation of SFKs [39]. It is worth mentioning that SFKs are also activated by other kinases, for example, Pyk2 and p125FAK, members of FAK family non-receptor tyrosine kinases, which are responsible for Src activation and recruitment [40,41].

### 2.2. Function of SFKs in Kidney

SFKs are of great significance in mediating signal transduction by interacting with various proteins and protein complexes [17,24]. The activation of SFKs is related to many regulatory signals from cell surface receptors, including growth factors, cytokines and immune cell receptors, G protein-coupled receptors, integrins, and other cell adhesion molecules [42,43]. When activated, SFKs trigger a cascade of intracellular signal transduction by phosphorylating specific tyrosine residues in other substrate proteins, such as STAT3, NF-κB, MAPK, and AKT [42,44,45]. SFKs have been shown to regulate a variety of cell functions, thus regulating a group of biological activities, including cell growth, survival, cytoskeleton remodeling, proliferation, and migration [43,46]. At the same time, SFKs have been implicated in the development of numerous diseases, including cancer [47], lupus nephritis [48], diabetes [49], Parkinson’s disease [50], and so on. The pathological function of SFKs in the kidney is shown in Table 1. 

SFKs are instrumental to the pathogenesis of kidney diseases and might be a promising target when it comes to the treatment of acute kidney injury. From this perspective, we pay more attention to the role and mechanism of SFKs in acute kidney injury.

## 3. The Pathophysiological Role of SFKs in AKI

### 3.1. Inflammation

Inflammation is a key factor in the occurrence and development of AKI, and it is also the core of the progression from AKI to CKD [66,67]. Changes in protein folding and mitochondrial function affect the innate immune response and ultimately lead to inflammation activation [6]. Of note, various cells, such as monocytes, macrophages, dendritic cells, and T cells [66,68], are involved in the development of AKI. Additionally, PRRs (Pattern recognition receptors), such as TLR (Toll-like receptor) and NLR (Nod-like receptor), can trigger activation of multiple kinases such as c-Jun N-terminal kinase (JNK), mitogen-activated protein kinase (MAPK), and nuclear factor kappa B (NF-κB), which result in the release of pro-inflammatory cytokines and chemokines, leading to the loss of function and apoptosis [66,69]. PRRs recognize conserved microbial structural units or pathogen-related molecular patterns (PAMPs), such as lipopolysaccharide (LPS), lipoteichoic acid, and porin [70] to orchestrate host defense against infection. PAMPs may directly interact with TLR and NLR expressed in renal parenchymal cells and resident immune cells [70,71] to induce inflammation in AKI.

Cytokine cascade reaction is first initiated by T cells, and then the release of cytokines by T cells triggers the activation of other immune cells, thus amplifying the cascade reaction [72]. Src, as a protein tyrosine kinase, can regulate NF-κB p65 and MAPKs, and is a vital molecule in the interrelated inflammatory cascade of kidneys in LPS-induced acute kidney injury [57]. Fisetin can inhibit this pathway, and its activation in the kidney of septic AKI mice shows anti-inflammatory and anti-apoptosis effects [57]. Similarly, another study suggests that Fisetin inhibits the macrophage-mediated inflammatory response by directly blocking Src and Syk [73]. The expression of inflammatory factors requires the activation of transcription factors [74]. The activation of NF-kB and STAT3 is of great significance for the occurrence and development of diseases such as AKI and acute pancreatitis [74,75,76]. It is suggested that Src may promote renal inflammation by activating STAT3 and NF-κB signaling pathways, and PP1 inhibits the expression of monocyte chemoattractant protein-1 in I/R-induced AKI and reduces macrophage infiltration [12]. Fyn aggravates renal fibrosis by promoting STAT3 phosphorylation, which indicates that Fyn can promote renal inflammation and fibrosis in the middle stage [61].

In our latest report, it was pointed out that Lyn can inhibit the activation of the NLRP3 inflammasome by phosphorylating NLRP3 in Tyr 918 and then promote its ubiquitination and proteasome-mediated degradation [77]. Consistent with this view, Lyn deficiency exacerbates lung inflammation induced by LPS, suggesting that Lyn plays a protective role in the acute lung injury model. However, the specific mechanism is unknown [78]. Some studies have found that Lyn can improve airway inflammation by inhibiting IL-13-induced NF-κB activity in airway epithelial cells in allergic inflammatory diseases [79]. Lyn overexpression decreases the phosphorylation of PI3K and Akt and inhibits ER stress in the lung, both of which could weaken the activation of NF-κB [79]. Moreover, Lyn negatively regulates the abnormal inflammatory response induced by Pseudomonas infection through SHIP-1 and IL-6/STAT3 signaling pathways [80].

The Signal-regulatory protein alpha (SIRPα) is an immune receptor mainly expressed on bone marrow leukocytes [81]. Interestingly, it has been found that, besides Lyn, SRC family kinase recruits SHP-1 by phosphorylating SIRPα, and SHP-1 controls the pro-inflammatory activation and expression of macrophages by inhibiting PI3K/AKT2 signaling cascade [81]. This study provides new insight into whether SFKs coordinate a fine-tuned synergistic regulation system through SIRPα to control the dynamic balance of inflammation.

In addition, many studies have clarified the role of SFKs in lupus nephritis. Studies have shown that Lyn and Fyn may be related to lupus nephritis, and mice lacking Lyn and Fyn exhibit severe kidney disease [48]. However, in the research of Sanae Ben Mkaddem et al., it was found that Fyn and Lyn may play different roles in maintaining homeostasis and inflammation in vivo. By phosphorylating SHP-1 on different residues, Lyn and Fyn show opposite regulatory effects on the ITAM receptor [62] (Figure 2).

### 3.2. Oxidative Stress

Reactive oxygen species (ROS) is produced during mitochondrial metabolism [82]. Low levels of ROS regulates cell signals, but excessive ROS induces oxidative stress (OS) [83]. Src family kinase inhibitor PP2 can improve oxidative stress in LPS-induced acute kidney injury [13]. ROS directly oxidizes c-Src and promotes the autophosphorylation of Tyr416, which leads to the enhancement of Src kinase activity [36,84]. Meanwhile, ROS production is also abolished with Src inhibition administration [85,86]. ROS-induced oxidative stress can activate the downstream Src/ERK1/2 signaling pathway, and Src and its downstream effector ERK1/2 are one of the most important upstream signals of apoptosis [87]. Orientin inhibits H2O2-induced apoptosis of PC12 cells by inhibiting ROS-mediated Src-MAPK/AKT signal transduction [88]. Src mediates the activation of STAT3 in vascular smooth muscle cells stimulated by Ang II, which increases ROS production and induces oxidative stress [89].

The Nuclear factor-E2-related factor 2 (Nrf2) is a transcription factor that plays a pivotal role in modulating antioxidant reactions [90]. Fyn is a negative regulator of Nrf2. Phosphorylated Fyn accumulates in the nucleus and activates the phosphorylation of Nrf2, which leads to the nuclear output, ubiquitination, and degradation of Nrf2, and then causes oxidative damage [91,92,93]. Pan and colleagues revealed that triptolide (TP) activates the GSK-3 β/Fyn pathway, promotes the cytoplasmic localization of Nrf2, and increases its subsequent degradation by the ubiquitin-proteasome pathway, which causes oxidative damage [64]. A recent study illustrated that the transcriptional function of Nrf2 activated by sulforaphane (SFN) is mediated by AMPKα2 through the Akt/GSK3β/Fyn pathway [60]. After SFN-induced AMPKα2 activation, AMPKα2 triggers Akt/GSK-3β phosphorylation and, as a consequence, prevents Fyn from entering the nucleus to output Nrf2, which leads to an increase in nuclear Nrf2 accumulation, thus weakening oxidative stress in type 2 diabetes [60]. In human lymphocytes, Green barley (GB) alleviates H_2_O_2_-induced oxidative stress by activating the Lyn/PI3K/Akt pathway [94] (Figure 3). 

### 3.3. ER Stress and Apoptosis

Endoplasmic reticulum (ER) stress is a physiological or pathological state that leads to the accumulation of unfolded or misfolded proteins in ER and plays an important role in maintaining protein homeostasis [59]. When endothelial or epithelial cells are stimulated, endoplasmic reticulum stress and subsequent unfolded protein response (UPR) can be induced. UPR may be adaptive and promote cell survival, or if endoplasmic reticulum stress is severe or long-term, it may lead to autophagy or cell apoptosis [6,95,96,97]. 

There are three transmembrane endoplasmic reticulum stress sensors: ERN1/IRE1 (endoplasmic reticulum to nucleus signaling 1), EIF2AK3/PERK (eukaryotic translation initiation factor 2-alpha kinase 3), and ATF6 (activating transcription factor 6) [98]. ERN1/IRE1 regulates XBP1 (X-box binding protein 1) processing and MAPK8/JNK1 (mitogen-activated protein kinase 8) activation, respectively [99,100,101]. Translocation of ATF6 to Golgi apparatus can drive the expression of endoplasmic reticulum chaperone protein and transcription factors XBP1 and CHOP [102]. Transcription factor CHOP is regarded as a prominent part of ER stress-induced apoptosis [102]. Active PERK can also directly or indirectly activate Nrf2 and transcription factor 4 (ATF4) [98]. The mechanistic target of rapamycin complex 1 (mTORC1) acts as a sensor and integrator for growth factors, amino acids, misfolded proteins in the ER, and the pressure-associated kinase eIF2 kinase [103]. mTORC1 mediates ER stress-induced apoptosis [104]. mTORC1 plays a key role in the pathogenesis of kidney disease.

Studies have shown that ROS activates upstream c-Src kinase and downstream mTOR to regulate endoplasmic reticulum stress in human proximal tubular cell line HK-2 [59]. Endothelial-mesenchymal transition (EndMT) is related to the development of fibrosis. ER stress-induced EndMT is mediated by Src kinase [105]. Under ER stress, Src is recruited to form a complex with IRE1α, which leads to the relocation of ER lumen chaperone protein on the cell surface [106]. In a murine model of renal ischemia/reperfusion and cisplatin-induced acute renal failure, Src kinase mediated apoptosis of renal tubular epithelial cells by activating ERK1/2 [12,107].

Overexpression of Fyn can activate the mTORC1 and IRE1α-JNK pathways at the same time, thus enhancing cell death induced by endoplasmic reticulum stress [108]. H_2_S improves the Akt/GSK-3 β/Fyn signal activated by ROS, thus increasing Nrf2 expression, which leads to the exertion of 20S proteasome function and further ameliorates uranium-induced ER stress-mediated kidney cell apoptosis [65].

The early phosphorylation of Akt may mediate the activation of mTORC1 by ER stress [79]. Overexpression of Lyn decreases the phosphorylation of PI3K and Akt and inhibits the activity of NF-κB, thus weakening endoplasmic reticulum stress [79]. This study also found that the levels of BIP and CHOP in mice with overexpression of Lyn decrease significantly [79] (Figure 4).

### 3.4. Autophagy

Autophagy is a defense mechanism against environmental stress, which is essential for cells to adapt to stress and maintain normal body balance [5]. It can eliminate damaged organelles and protein aggregates and maintain cell homeostasis [109,110]. However, excessive or insufficient autophagy also exerts damage to cells [5]. Numerous studies have shown that autophagy protects cells from cell death in AKI through various mechanisms [111].

Mitochondrial breakage and severe oxidative stress also engage in inducing autophagy [112]. Amino acids regulate autophagy by activating mTORC1 [113]. Src inhibits autophagy by promoting the dissociation of GATOR1 from Rags and mediating the recruitment and activation of mTORC1 induced by amino acids on the lysosomal surface [114]. Oxidative stress induced by NADPH oxidase 2 (Nox2) stimulates the activation of Src kinase and then inhibits mTOR-dependent autophagy through PI3K/Akt/mTOR pathway [115]. ER stress is involved in inducing autophagy [99]. Soo Young Moon et al. found that ER stress by tunicamycin (TM) and toxic carotene (TG) induced EMT through autophagy by activating c-Src kinase in renal tubular epithelial cells [116].

AMP-dependent protein kinase (AMPK) also plays a key role in autophagy [117]. Studies have shown that Fyn gene deletion increases AMPK activity depending on LKB1 regulation [118,119]. Consistent with this view, A study has found that Fyn inhibits AMPK through the LKB1 and PIKE-A pathways [120]. Recent reports have suggested that inflammatory factor TNF-α activates Fyn kinase, and Fyn specifically phosphorylates AMPKα on Y436, inhibiting AMPK activity and thus inhibiting autophagy [121]. Fyn inhibits AMPK activation and increases mTORC1 activity, but recent studies have shown that Fyn can also regulate autophagy through the AMPK/mTORC1 independent pathway [122,123]. Fyn inhibits macroautophagy by reducing Vps34 protein level in a STAT3-dependent manner and then reducing Vps34/p150/Beclin1/Atg14 complex [122].

The positive regulation between Lyn and autophagy has been reported [124,125], but little research has been done. Lyn was found to promote cell survival by promoting autophagy in nutrient-deficient glioblastoma cells [124]. Lyn may act as a bridge between TLR2 and autophagy after Pseudomonas aeruginosa infection [125]. TLR2 initiates phagocytosis and activates Lyn, which promotes the recruitment of LC3, regulates autophagy through Rab and cofilin, and mediates the fusion of lysosomes with autophagy containing Pseudomonas aeruginosa to promote autophagy [125]. Contrary to the above studies, Lyn kinase inhibits apoptosis and autophagy through the PI3K/AKT signaling pathway in melanoma cells [126]. Whether the difference between the two studies is due to the different roles of Lyn in different cells is unknown and needs further study, but other studies suggest that it is possible for Lyn to play a positive role in autophagy (Figure 5).

### 3.5. Fibrosis

As we all know, severe kidney injury, even mild acute injury with persistent inflammation, is likely to develop into renal fibrosis [6,127,128]. Therefore, this topic has attracted wide attention. Unfortunately, the exact mechanism of CKD progression after AKI is still unknown, but now, emerging evidence shows that maladjustment repair is the main cause of renal fibrosis and AKI-induced CKD [4]. This is the result of the interaction of cell death, endothelial dysfunction, senescence of renal tubular epithelial cells, and inflammatory process [9]. The hypothesis of endothelial-interstitial transition (EndMT) and epithelial-interstitial transition (EMT) has been put forward, and it is believed that they are involved in renal fibrosis [4,9,129,130]. Renal fibrosis is characterized by the accumulation of extracellular matrix (ECM) due to activation and proliferation of myofibroblasts, which leads to kidney damage [9].

Emerging evidence about the role of Src kinase in renal fibrosis has been reported in recent years [38], but the underlying mechanisms remain not fully understood. Previous studies demonstrated that severe AKI causes proximal tubular epithelium cells to produce and secrete fibrotic factors, such as transforming growth factor β (TGF β) and connective tissue growth factor (CTGF) [131,132]. TGF-β 1 is one of the most important fibrotic factors in renal fibrosis, which can induce myofibroblast transformation [133] and EMT to increase extracellular matrix [134]. Src is activated during the development of renal fibrosis induced by UUO injury, which leads to TGF-β 1-induced activation and proliferation of renal interstitial fibroblasts and accumulation of ECM [51]. Src kinase inhibitors prevent the renal epithelial cell cycle from arresting and ameliorating kidney fibrosis after UUO injury [51]. In accordance with this view, Chen et al. found that Src kinases induce the phosphorylation and activation of epidermal growth factor receptor (EGFR), leading to TGF-β 1-mediated fibrosis [52]. In addition, during the early stage of renal interstitial fibrosis, the activity of matrix metalloproteinase-2 (MMP-2) is increased, which promotes the interstitial transition of renal tubular epithelial cells, and leads to the production of ECM [135,136]. It is found that renal hypoxia can activate Src and reduce MMP-2 activity, which further aggravates renal interstitial fibrosis [37]. In NRK-52E cells and senile male Fisher rats, Src kinase inhibitor PP2 is able to suppress the up-regulation of matrix metalloproteinase (MMP)-7 and reduce the expression of collagen Col1a2, which attenuates collagen deposition in the kidney [137]. Apart from PP2, tamoxifen may also be beneficial in the treatment of renal fibrosis by inhibiting Src kinase [138]. In the latest research report, dasatinib alleviates renal fibrosis by inhibiting Src, c-Abl, STAT-3, and NF-κ B signaling in the UUO-induced renal fibrosis model [53].

Moreover, Na/K-ATPase binds to Src to form a complex that keeps Src inactive, which is essential in the pathogenesis of renal injury and fibrosis [139,140]. As a derivative peptide of Na/K-ATPase, pNaKtide also inhibits the activity of Src [140,141], leading to the decrease of myofibroblast accumulation and ECM deposition, the down-regulation of TGF-β1 expression, and the improvement of renal fibrosis [140]. KIM et al. demonstrated that the Farnesol X receptor (FXR) plays a key role in preventing fibrosis as the excitation of FXR inhibits the activity of Src [54]. It is found that FXR can regulate renal fibrosis through the FXR-Src-YAP pathway [54]. These studies confirm the notion that Src is indispensable in the process of renal fibrosis.

Recent studies found that Fyn, a member of the SFKs family, is also engaged in mediating kidney fibrosis, and the expression of Fyn is up-regulated in the UUO-induced renal fibrosis mouse model [61]. Src family kinase inhibitors (SU6656 and PP2) reduce the deposition of ECM protein stimulated by TGF-β. The expression of E-cadherin is decreased in the UUO-induced model in Fyn-/-mice [61]. Interestingly, unlike PP1 inhibiting Src to block Smad3 and STAT3 signaling pathways [51], Fyn-mediated renal fibrosis is mediated by the non-Smad signaling pathway, that is, caused by activation of STAT3, and Smad3 and AMPK signaling transduction are not essential in this process [61]. Inhibition of Fyn restrains EGFR and Akt signaling, which indicates that the lack of Fyn may be related to these mechanisms independent of the STAT3 cascade [61].

Wei et al. showed that Hck is also a key SFKs member involved in renal fibrosis [142]. In renal tubular cells, overexpression of Hck activates TGF-β/Smad signaling, while dasatinib treatment in UUO kidney of mice reduces phosphorylation of Smad3 after inhibiting Hck [142]. In addition, in the animal models of lupus nephritis and folic acid nephropathy, it is found that dasatinib administration reduces the expression of fibrosis markers such as Col1a1, fibronectin, and vimentin, which further attenuates the progression of fibrosis [142] (Figure 6).

## 4. Targeting SFKs for AKI

Today, the approved drugs for treating AKI are still very limited. In view of the possible role of SFKs in AKI, a large number of studies are exploring the strategies of SFKs small molecule inhibitors for AKI. The effects of SFKs inhibitors on the pathophysiology of AKI are summarized in Table 2.

For LPS-induced AKI, PP2 treatment is effective as it reduces inflammatory and oxidative stress [13]. At the same time, mitochondrial biogenesis is improved [13]. Additionally, PP2 is a sort of cure for renal fibrosis, as many studies have proved that PP2 can inhibit Src to treat renal fibrosis. [51,54,137]. KF-1607, a newly synthesized Src kinase inhibitor with low toxicity, has a similar effect compared to PP2 and can inhibit renal inflammation and oxidative stress and prevent the development of tubulointerstitial fibrosis in UUO mice [143].

PP1 can inhibit the phosphorylation of Src kinase as it is a highly selective Src kinase family inhibitor [149]. PP1 treatment inhibits the phosphorylation of NF-κB and STAT3, and it also suppresses the expression of neutrophil gelatinase-related lipid transport protein and macrophage infiltration in the kidney [12]. In addition, PP1 reduces the apoptosis of damaged kidneys by inhibiting ERK1/2 phosphorylation and maintains the integrity of adhesion and tight junctions in renal epithelial cells [12]. In human mesangial cells (HRMC), LPS induces VCAM-1 expression through c-Src, and increases monocyte adhesion and inflammatory reaction, while PP1 disturbs this process [85]. PP1 and PP2 mitigate the activation of PKC δ induced by cisplatin, thus relieving renal cell injury and the nephrotoxicity of cisplatin [144]. Pretreatment with PP1 inhibitor significantly inhibits caspase-3 activation induced by cisplatin and improves cell morphology and apoptosis in an Src/ERK-dependent manner [145].

Dasatinib is a novel multi-target inhibitor that can effectively inhibit Abl and SFKs [150]. Dasatinib can not only block the cytotoxicity of T cells but also quickly and completely shut down the release of inflammatory factors such as IL-6, TNF-α, and IL-1β [146]. More importantly, dasatinib also quickly prevents the activation of cytokine release signals in T cells [146]. In addition, in the UUO model, dasatinib treatment decreases the expression of inflammatory markers (CCL3, CCL5, TNF α, IL-1 β, and MCP-1) and inflammatory macrophage infiltration [53,142], down-regulates renal expression of α-SMA and fibronectin [53], and alleviates renal oxidative stress, inflammation and fibrosis [53,142]. All the studies suggested the notion that dasatinib elicits anti-inflammation and anti-fibrotic potency in animal models.

Nintedanib is a triple kinase inhibitor that has been widely studied in pulmonary fibrosis and has been approved for the treatment of idiopathic pulmonary fibrosis [15]. Liu et al. recently studied the role of nintedanib in renal fibrosis and found that nintedanib inhibits the phosphorylation of Src, Lck, and Lyn of SFKs [147]. It suppresses inflammatory reaction and macrophage infiltration, restrains the activation of renal interstitial fibroblasts, reduces the deposition of ECM in the kidney, and even reverses renal fibrosis [147]. They also indicated that the combination of nintedanib with gefitinib in the UUO model of mice exerts greater anti-fibrosis effects [148].

Src is the most studied kinase in SFKs, while other members, including Fyn and Lyn kinases, are hardly studied in AKI, so other members may have a broad research space. However, some studies have found that both total Fyn and phosphorylated Fyn increase in LPS-induced AKI [67], but the specific mechanism is unknown.

## 5. Conclusions

Acute kidney injury is the main culprit of death. Oxidative stress, inflammation, endoplasmic reticulum stress, autophagy, apoptosis, and fibrosis are vital pathogenesis of AKI and AKI progression to CKD.

SFKs are considered the essential mediator in modulating signal transduction. Specific inhibitors of SFKs have proven to be available in in vitro and in animal models, and the application of small molecule inhibitors targeting these kinases is expected to achieve the therapeutic effect of AKI. This paper mainly summarizes the functions of SFKs members Src, Fyn, and Lyn, but the details require further study. First, the role of SFKs in renal pathophysiology has aroused great interest, but its role in AKI has not been clarified completely. Secondly, from the current research, it is found that Src and Fyn generally play an active role in the pathogenesis related to AKI in kidney and other diseases, while Lyn kinase, contrary to Src and Fyn, plays a negative role in inflammation, oxidative stress, autophagy, and apoptosis, and can alleviate inflammation and apoptosis and promote autophagy (Figure 7). However, whether they have opposite mechanisms in the same disease needs further study. 

## 6. Future Perspectives

At present, there are still three obstacles to determining the role of SFKs in AKI. First, 11 SFK members have been identified so far, but only some of them, such as Src, Fyn, and Lyn, have been discussed regarding their role in kidney diseases. Second, multiple members of SFKs are expressed in specific cells or tissues and play their different roles. Therefore, silencing a single kinase may not be enough to prevent the influence of pathophysiological development. Third, SFKs inhibitors do not target a specific member of kinase, so it is difficult to clarify the role of a single member of SFKs in specific diseases. Therefore, future research should be conducted to elucidate the role of SFKs by using highly selective inhibitors and gene knockout techniques. In addition, the analysis of the expression profile of SFKs in renal biopsy tissues will also help to clarify the role of SFKs in AKI. Although the exact mechanism of SFKs in AKI remains to be clarified, SFKs may become a potentially new therapeutic target for AKI.

## Figures and Tables

**Figure 1 biomolecules-12-00984-f001:**
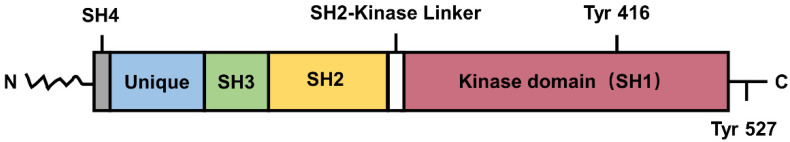
The domain structure of Src family kinases. The chicken numbering system is displayed. 1 SFKs consist of several fields: the SH4 domain (in gray), a unique domain (in light blue), the SH3 domain (in green), the SH2 domain (in orange), and the SH1 domain (in red).

**Figure 2 biomolecules-12-00984-f002:**
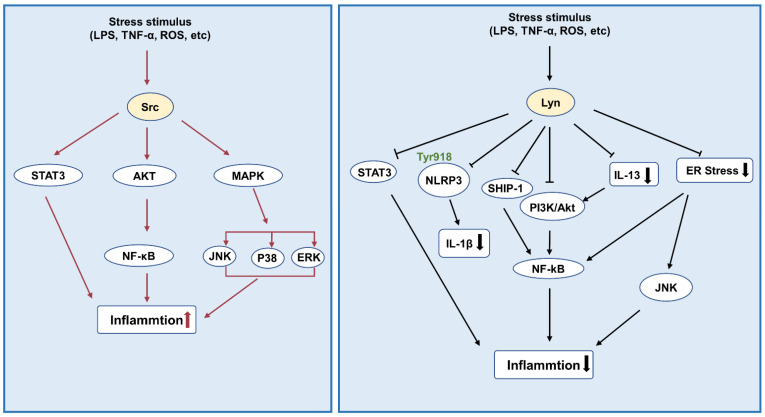
Src family kinases and inflammation. Src activates STAT3, NF-κB, and MAPK signaling to promote inflammation, while Lyn kinase inhibits the activation of STAT3, NF-κB, and PI3K/Akt pathways to alleviate inflammation. Abbreviations: ROS: reactive oxygen species; ERK: extracellular signal-regulated kinase; NLRP3: pyrin domain-containing 3 protein; SHP-1: Src homology 2 domain-containing protein tyrosine phosphatase 1; PI3K: phosphatidylinositol 3-kinase; AKT: also known as protein kinase B (PKB).

**Figure 3 biomolecules-12-00984-f003:**
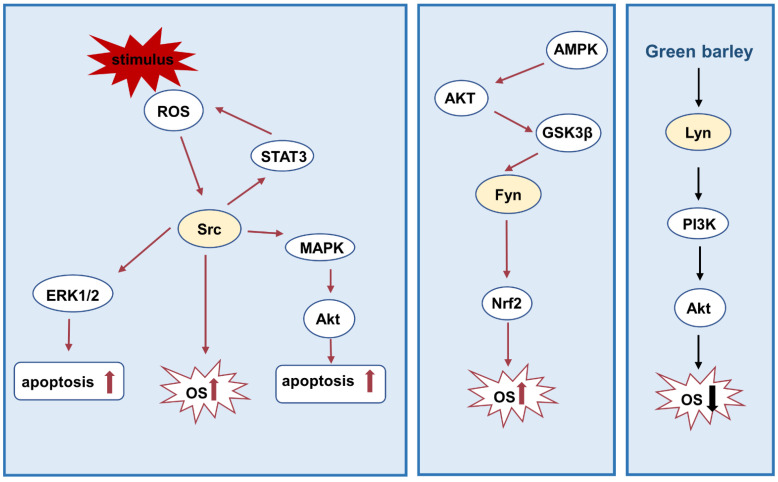
Src family kinases and oxidative stress. Src kinase triggers STAT3, MAPK, Akt signaling pathway activation under ROS stimuli, thus promotes oxidative stress and apoptosis. Fyn promotes oxidative stress mainly by Nrf2. On the contrary, the activation of Lyn alleviates oxidative stress through the PI3K/Akt pathway. Abbreviations: GSK-3β: glycogen synthase kinase-3beta.

**Figure 4 biomolecules-12-00984-f004:**
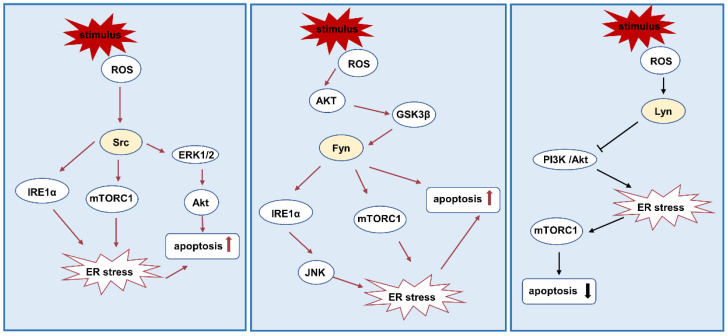
Src family kinases and ER stress. Src and Fyn aggravate oxidative stress and apoptosis by mediating mTORC and ERK1/2 and also interact with IRE1α to cause oxidative stress. Lyn activation reduces oxidative stress and apoptosis through mTORC. Abbreviations: IRE1α: inositol-requiring enzyme 1alpha.

**Figure 5 biomolecules-12-00984-f005:**
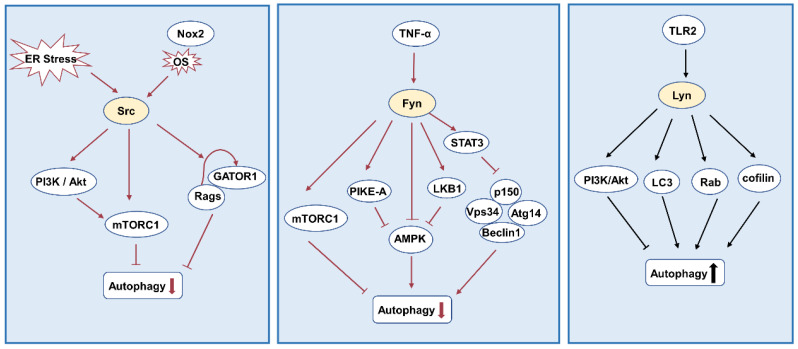
Src family kinases and autophagy. Src and Fyn inhibit autophagy through mTORC. Fyn also reduces autophagy by inhibiting AMPK phosphorylation. Lyn may have a two-sided effect on autophagy. Abbreviations: GATOR1: gap activity toward rags 1; Rags: Ras-related GTPases; LKB1: liver kinase B1; LC3: light chain 3.

**Figure 6 biomolecules-12-00984-f006:**
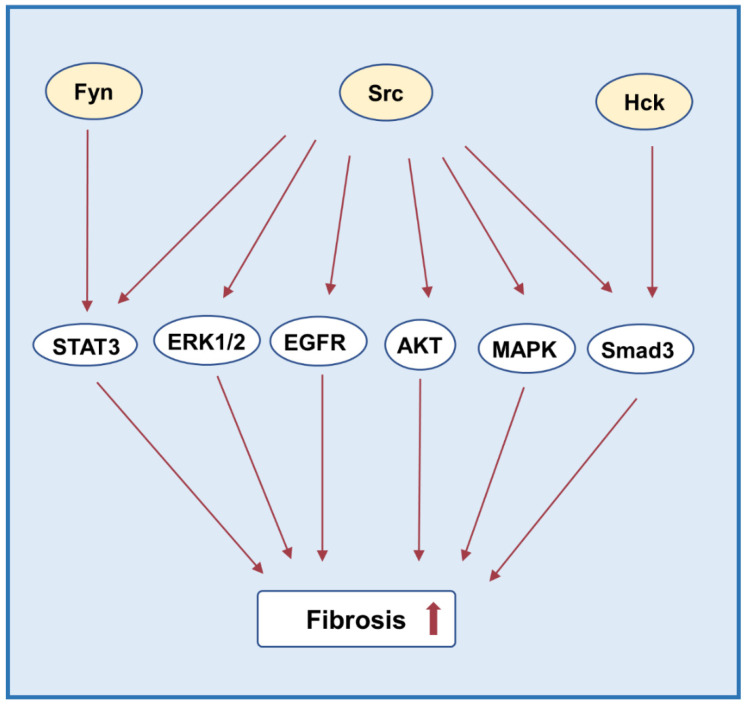
Src family kinases and fibrosis. Src activation results in phosphorylation of signal proteins STAT3, AKT, MAPK, and EGFR. Src also promotes the activation of Smad3 and ERK1/2. Fyn and Lck activate STAT3 and Smad3, respectively.

**Figure 7 biomolecules-12-00984-f007:**
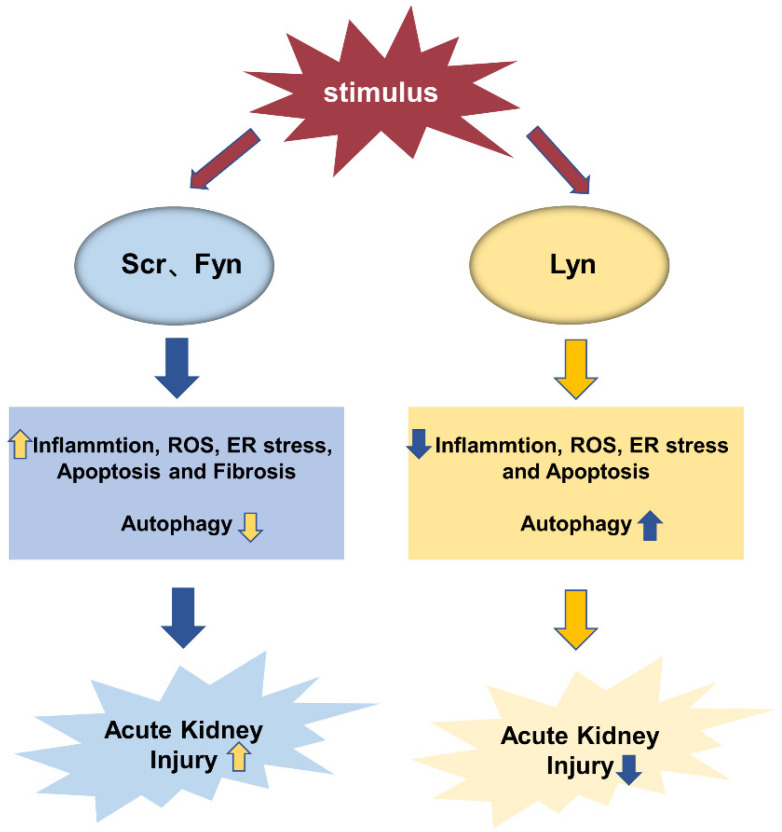
The role of SFKs in AKI.

**Table 1 biomolecules-12-00984-t001:** The pathological function of SFKs in the kidney.

Member of SFKs	Organs/Cells	Models	Mechanisms	References
Scr	Kidney	renal ischemia/reperfusion model	Reduces renal injury by activating STAT3, ERK1/2, and NF-κB signaling pathway	[12]
	Kidney	unilateral ureteral obstruction	Mediates the activation of TGF-β1 signal, NF-κB, Smad-3 epidermal growth factor receptor and STAT3, and EGFR transactivation	[51,52,53]
	Kidney	unilateral ureteral obstruction	Regulates phosphorylation and localization of YAP	[54]
	Kidney	STZ-induced type 1 diabetes	Mediates phosphorylation of EGFR and MAPK	[55]
	Kidney	diabetic db/db	Induces activation of p38 MAPK activation	[56]
	Kidney	LPS-induced septic AKI	Mediates NF-κB and MAPK signaling pathways	[57]
	podocytes	HIV-associated nephropathy (HIVAN)	Activates of STAT3 and MAPK1, 2 Mediates cell proliferation and dedifferentiation of podocytes	[58]
	HK-2	hypoxia	Decreases MMP-2 activity and aggravates renal interstitial fibrosis	[37]
	HK-2	ER stress	Activates mTOR pathway	[59]
Fyn	Kidney	STZ-induced type 1 diabetes	Suppresses Nrf2 expression	[49]
	Kidney	type 2 diabetes-induced nephropathy	Promotes the output of Nrf2 from nucleus	[60]
	Kidney	obstructive fibrosis	Mediates STAT3 activation	[61]
	Kidney	lupus nephritis	Mediates ITAM phosphorylation to promote inflammation	[62]
	Podocytes	apoptosis	Activates of Fyn-induced TRPC6 phosphorylation	[63]
	NRK-52E	oxidative stress	Mediates degradation of Nrf2	[64,65]
Lyn	Kidney	lupus nephritis	Mediates ITAMi phosphorylation to homeostasis	[62]

Abbreviations: STAT3: signal transducer and activator of transcription 3; NF-kB: the nuclear factor kappa B; MAPK: mitogen-activated protein kinases; Akt: also known as protein kinase B; TGF-β1: transforming growth factor beta1; Nrf2: nuclear factor E2-related factor 2; ITIM: immunoreceptor tyrosine-based inhibitory motif; ITAMi: inhibitory immunoreceptor tyrosine-based activation motif; TRPC6: transient receptor potential cation channel C6; ERK1/2: extracellular signal-regulated kinases 1 and 2; mTOR: mechanistic target of rapamycin; ER: endoplasmic reticulum; NRK-52E: renal proximal tubular cells; HK-2: human proximal tubular cells.

**Table 2 biomolecules-12-00984-t002:** Effect of targeting SFKs inhibitors on the pathophysiology of AKI.

Compounds	Targeted SFKs	Effects	Reference
PP2	Src/Fyn	Improves mitochondrial dysfunction and renal injury induced by LPS,	[13]
	Src	Reduces collagen deposition and improves fibrosis in kidney	[137]
KF-1607	Src	Inhibits renal inflammation and oxidative stress, prevents tubulointerstitial fibrosis	[143]
PP1	Src	Relieves renal injury in mouse model of renal ischemia/reperfusion (I/R)	[12]
	Src	Reduces the expression of VCAM-1 in human mesangial cells (HRMC) treated with LPS and alleviates monocyte adhesion and inflammatory reaction	[85]
	Src	Reduces the damage and death of renal cells induced by cisplatin	[144]
	Src	Inhibits apoptosis after cisplatin treatment by Src/ERK signaling pathway	[145]
	Src	Inhibits the activation and proliferation of renal interstitial fibroblasts, regulates the expression of cyclin, and improves fibrosis	[51]
dasatinib	Src/lck/Hck/c-Abl	Decreases inflammatory macrophage infiltration and renal oxidative stress, reduces renal expression of α-SMA and fibronectin, and improves fibrosis	[53,142,146]
nintedanib	Src/Lck/Lyn	Inhibits inflammation and renal fibrosis	[147,148]

## Data Availability

Not applicable.

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
