# Peer review of "Src Family Kinases: A Potential Therapeutic Target for Acute Kidney Injury"

_biomolecules, 2022, doi:10.3390/biom12070984_

Round 1

Reviewer 1 Report

The authors give an overview about the role of Src family kinases and their role in acute kidney injury. The potential to use Src family kinases as pharmaceutical target in AKI is very interesting and the review is giving a great overview of the topic.

Major

In chapter 2 the description of SFK family members is misleading. To my knowledge Yrk (Yes related kinase) was discovered in chicken and never found in humans. The authors should make that clear. Furthermore, the addition of Srms, Brk and Frk to the family should be stated that these kinases are also rather a-typical family members. They are mostly grouped in a separate family, the Brk family (Brauer et al 2010) and all these kinases are not myristoylated.

Additionally, the SH4 domain is not only important for the localization. It is also important for the regulation of the kinase by a binding mechanism to the SH 1 domain (Ahler et al 2019, PMID: 30956043).

In Page 2, Line 67 the authors refer to Tyr530. It is unclear which kinase the authors refer to. Since the absolute position of the phosphorylation sites is changing with the different kinases and species. Tyrosine Y416 is referred to the chicken Src kinase, which would correspond to Y527 in this kinase. The authors should clarify that.

Figure 1 should entail where the figure is adapted from. Furthermore, in the Figure description, the color code and the domains should be described. Again, here the refereeing to Tyr530 is wrong in the context with Tyr416. It should be Tyr527 for chicken Src. For human Src the authors should useTyr 419 and Tyr530.

Page 2 line 56 needs more references, it is only referred to a BRK discovery paper. There should be the manuscripts cited for Frk and Srms.

Page 2, line 67 references for the activation of Src family kinases are not sufficient and misleading. Like there is only the structure of Src from 1997 and the activation by hypoxia. Please refer to other papers. Suggestions for additions would be e.g. Berndt et al 2021 PMID: 34204297, Luttrell et al, Satoh et al 2005, Lin et al 2007, Ramnath et al 2009 …

In page 2 line 75 the authors referencing a paper which shows the discovery of BRK from 1994, whereas it is talked about the interaction partners of SFKs. In this line the authors say that the kinases are cytoplasmic proteins, nevertheless they are membrane localized.

In Figure 1 is the placement of the inhibitors not clear. PP1 is an inhibitor shown to inhibit all SFKs and the authors just placed with Src and not Lyn. Is there a reason for this? Furthermore, Dasatinib is also inhibitin Src and is shown to block only the stress stimulation.

Also in Figure 3, the authors chose to only place the PP2 inhibitor with Src, whereas it is also inhibiting Fyn and Lyn. Could this be corrected or is there a reasoning?

For both figure legends (Figure 2 and 3), the description should be a more detailed.

Table 1 is misleading, due to the earlier stated fact that PP1, PP2 and Dasatinib inhibit all members of SFKs. Again, the formatting is not ideal for the table. It would be clearer if the effects would be separated with specific lines.

Throughout the text it is stated that PP1 is a highly selective Src inhibitor (page 11 line 276). This is not true, PP1 is a reversible Src family kinase inhibitor (Src, Lck, Fyn, Hck …) which is also able inhibiting EGFR and JAK2 kinase and others through a weaker interaction.

The authors state that Dasatinib is only inhibiting Src and Lck kinase. Multiple studies show that Dasatinib can inhibit Lyn, Fyn and other family members. Could this be corrected

Furthermore, there should be a reference included on Page 11 line 283.

Minor:

Page 1, Line 16: abbreviation of AKI without explaining what it stands for.

Page 1, Line 17: Space between oxidative stress and (OS) as well as endoplasmic reticulum and (ER)

Page1, Line 39: before NO two spaces

Page 1, Line 42: space before (CKD)

Page 2, Line 67: And The … and the

Page 2, Line 71: Formatting (Figure 1)

Page 3 – 5: formatting of the table. It is unclear which mechanism belongs to which kinase. There should be more space or a line in between.

Page 10, Line 242: Formatting of the brackets

Page 11, Line 273: TABLE 2 instead of TABLE 1

Reviewer 2 Report

This review article by Li and colleagues discusses the therapeutic potential of targeting the Src Family kinases in the treatment of acute kidney injury. Whilst the review in general is well-written, it requires better integration of the various sections discussed, and further detail in certain areas – see comments below:

i) There needs to be a better link between the 1. Introduction and 2. Overview of SFKs sections i.e in Section 1., the Authors should outline the limitations of currently-available treatments for AKI and why currently identified therapeutic targets have not provided adequate protection of disease progression (particularly when treatments are administered to established disease pathology). This should then lead onto trying to identify new therapeutic targets that may provided broader protection of AKI, such as the SFKs.

ii) The sentence on lines 95-96 should be revised to read as: ‘SFKs are instrumental to the pathogenesis of kidney diseases and might be a promising target….’

iii) Fibrosis is a prominent feature of AKI – hence, the role of SFKs in the progression of 3.5 Fibrosis should be discussed, as per what is included for 3.1 Inflammation; 3.2 Oxidative stress; 3.3 ER stress and apoptosis; and 3.4 Autophagy.

iv) Section 4. Targeting of SFKs for AKI is too brief and should be expanded. A quick search in PubMed of ‘PP1 and Src and kidney’ revealed 40 publications / ‘PP2 and Src and kidney’ revealed 111 publications / ‘Datasinib and kidney’ revealed 67 publications / ‘Fisetin and kidney’ revealed 42 publications – yet the Authors have only included a few lines on the effects of these compounds. It would be helpful to readers of this review to have a better insight into the detailed mechanism(s) of actions by which these compounds induce their therapeutic effects in the kidney/as a treatment for AKI.

Round 2

Reviewer 1 Report

Dear Authors,

Thank you for addressing all my suggestions. 

Author Response

It’s our pleasure to have this opportunity. Thank you!

Reviewer 2 Report

The Authors have addressed the main issues raised on their original submission, but in doing so need to improve the grammatical content of the new sections they have added, and the revised review article in general. 

Author Response

Thank you for your genuine and professional advice and suggestions. We have revised the latest part and corrected the grammar mistakes in the whole article. If there are any questions, please let us know. Thank you again!